# A national cross-sectional study of the role of clinician specialty and facility complexity on glucocorticoid prescribing in Veterans

Beth I. Wallace [1,2,3] ✉, Shirley Cohen-Mekelburg[1,2,3], Tony Van[1], Rachel Lipson[1], Brooke Kenney[4], Chelsea Tatchin[4], Jessica Baker[1], Sameer D. Saini[1,2,3] & Akbar K. Waljee[1,2,5]

## Abstract

**Background:** Glucocorticoids are used commonly despite their toxicity and availability of effective alternatives. Prior claims-based studies evaluating effect of clinician specialty on glucocorticoid prescribing do not examine how facility-level resources affect prescribing patterns. We aim to describe the effect of facility complexity and clinician specialty on oral glucocorticoid prescribing in the general population.

**Methods:** This is a national cross-sectional study of Veterans prescribed oral glucocorticoids during 2021-2022. We defined glucocorticoid use as ≥1 outpatient pharmacy claim for oral glucocorticoids, and prolonged use as ≥30 days' supply dispensed within 365 days. We defined complexity, rurality, and region using VHA operational metrics, and clinician specialty using NUCC taxonomy codes. We descriptively evaluated demographics, comorbidities, and healthcare utilization stratified by glucocorticoid use. We calculated the proportion of users and prolonged users at each facility, stratified by complexity, rurality, and region. We calculated the proportion of glucocorticoid prescriptions by clinician specialty. For three high-prescribing specialties, we calculated the facility-level proportion of glucocorticoid prescriptions by complexity designation.

**Results:** Here we show that among 5,729,134 patients at 124 facilities, a mean of 12.1% (SD 3.5%) are glucocorticoid users; 2.1% (SD 0.5%) were prolonged users. Rates of use and prescribing vary substantially within and across facility complexity designations. Family practice represents 18.8% of glucocorticoid-prescribing clinicians but prescribes 26.3% of filled prescriptions. Family practice displays higher prescribing at lower-complexity sites.

**Conclusions:** In this large national population, overall and prolonged glucocorticoid use are common and prescribing patterns vary by clinician type and complexity designation. Clinician-facing interventions to address knowledge gaps, incentivize non-glucocorticoid treatments, or facilitate specialty care access may reduce overuse among high-prescribing clinicians.

## Plain language summary

Glucocorticoids are commonly used steroids to treat inflammatory and autoimmune conditions. This class of medication has downsides like toxicity and alternative medications exist that could be used in their place. We evaluated prescribing patterns based on clinician specialty and facility characteristics for US Veterans. In a national population of over 5 million Americans receiving medical care in the Veterans Health Administration during 2021-2022, we found that one in eight received a glucocorticoid prescription and one in 50 used glucocorticoids for a prolonged period of time. We found substantial differences in rates of glucocorticoid use within and across facilities with different levels of complexity, and in rates of prescribing within and across provider specialty. This highlights the need for interventions to reduce glucocorticoid overuse among high prescribing clinicians and at facilities that use this medication frequently.

Glucocorticoid steroids, like prednisone, are commonly used to treat acute, self-limited conditions like bronchitis, sinusitis, and allergic skin conditions, even when evidence of benefit is lacking[1,2]. Most patients receive one to two week course of glucocorticoids, which have long been considered safe due to the short exposure duration[3,4]. However, growing evidence suggests even short-term glucocorticoid exposure is associated with substantial population-level risk of heart failure, venous thrombosis, sepsis, and gastrointestinal bleeding[1,5]. Glucocorticoid use also increases the risk of death in patients with mild to moderate COVID-19, despite its efficacy in treating severe disease[6–8]. These adverse effects, and others such as osteoporosis, cataracts, diabetes, and

fungal infections, are even higher among recipients of prolonged glucocorticoids, especially older adults and those with comorbidities predisposing them to glucocorticoid toxicity[9]. Glucocorticoid use is highly sensitive to clinician preferences, with high-prescribing clinicians using glucocorticoids as much as 64% more often than low-prescribing clinicians[10].

Stakeholders across different specialties have proposed a "glucocorticoid stewardship" model similar to that used for antibiotic and opiate prescribing[11–13]. Under this model, targeted interventions could be developed to reduce inappropriate prescribing (i.e., when net harm is likely and/or glucocorticoid-sparing options exist). However, we must first understand which patients are receiving glucocorticoids and how clinicians are prescribing them. There is limited and variable data characterizing glucocorticoid use and prescribing in the general population of the United States (US). An analysis using National Health and Nutrition Examination Survey (NHANES) data between 1999 and 2008 reported the prevalence of glucocorticoid use at 1.2%. An analysis of Optum claims data between 2009–2018 reported an average annual prevalence of 6.8%, increasing over time to 7.7%[14,15]. In the same study, family practice physicians accounted for 31.6% of overall glucocorticoid prescribing. However, these studies do not account for use and prescribing patterns that reflect newly recognized risks related to COVID-19, do not capture pharmacy dispensing data that more accurately reflect exposure, and do not adequately reflect elderly and/or underinsured populations who are more likely to use glucocorticoids in lieu of more expensive glucocorticoid-sparing treatments[1,14]. Existing studies also do not examine how differences in facility-level resources affect these patterns. Facility-level variation in glucocorticoid prescribing is critical to understanding how clinicians who practice in resource-rich settings with information-sharing and easy access to specialty care might prescribe very differently than clinicians without these resources[16].

This study aims to examine the prevalence of oral glucocorticoid use and patterns of clinician prescribing in a national population of US Veterans between 2021 and 2022. Specifically, we build on previous studies by evaluating how clinician specialty and facility complexity designation affect oral glucocorticoid use and prescribing patterns among Veterans who received glucocorticoid treatment. We find that, among 5,729,134 patients at 124 facilities, a mean of 12.1% (SD 3.5%) are glucocorticoid users; 2.1% (SD 0.5%) were prolonged users. Rates of use and prescribing vary substantially within and across facility complexity designations. Family practice represents 18.8% of glucocorticoid-prescribing clinicians but prescribes 26.3% of filled prescriptions. Family practice displays higher prescribing at lower-complexity sites.

## Methods
### Inclusion and ethics
The study was conducted in accordance with the Declaration of Helsinki. The study plan was approved by the Institutional Review Board of the VA Ann Arbor Healthcare System, ID 1616863.

### Study setting
This was a national retrospective cross-sectional study of US Veterans ≥18 years receiving care at Veterans Health Administration (VHA) facilities between 1 January 2021 and 31 December 2022. VHA is the largest integrated health system in the US, providing care to over nine million Veterans at 1138 outpatient care sites[17]. Inclusion required ≥1 VHA outpatient medical claim and ≥1 outpatient pharmacy claim between 2021–2022. We excluded Veterans without a VHA primary care provider (PCP), as they were likely to receive a substantial portion of medical care in civilian health systems[18]. We also excluded Veterans with PCPs located at a facility ($N = 5$) who transitioned to the Cerner electronic medical record system during 2021 or 2022, as data post-transition is unavailable in the Corporate Data Warehouse (CDW) claims database.

Patient demographics, diagnosis codes, and healthcare utilization data were extracted from medical claims, and oral glucocorticoid prescriptions were extracted from pharmacy claims. We limited our evaluation to oral

glucocorticoids given limited data on the systemic absorption of inhaled and topical glucocorticoids, and limitations of claims data for characterizing exposure to intravenous and intramuscular glucocorticoids. Comorbidity data was categorized using the Healthcare Cost and Utilization Project Clinical Classifications Software Refined. Comorbidities were defined as present if a diagnosis code was present during the study period. All patients with ≥1 outpatient oral glucocorticoid prescription dispensed between 2021–2022 were classified as glucocorticoid users (Supplementary Table 1). The remainder were classified as non-users. All glucocorticoid users with ≥30 days' supply of oral glucocorticoids dispensed in a 365-day period were classified as prolonged users[4]. Intravenous glucocorticoids were excluded as they are used for a small subset of highly specific indications and would not provide information relevant to the general population.

Facility complexity designation was classified into five categories (1a, 1b, 1c, 2, 3 in descending order) based on patient volume and characteristics, scope of clinical services offered, education and research costs, and administrative complexity[19]. A facility with a high complexity designation, e.g., level 1a or 1b, is more resource-rich than a lower complexity facility. We assigned geographic region (Continental, Midwest, North Atlantic, Southeast, or Pacific) using an operational VHA designation that groups facilities based on Veterans Affairs Integrated Service Network catchment areas[20]. Facilities were classified as Urban or Rural based on the Rural-Urban Commuting Area Codes framework used operationally by VHA[21].

We identified glucocorticoid prescribing clinicians by linking prescription claims with clinician security identifiers (SIDs) available in CDW. To determine clinician specialty, each clinician's SID was linked to their national clinician identifier (NPI), a standardized, publicly available identifier assigned by the Centers for Medicaid and Medicare Services (CMS). Each NPI was then cross-referenced with the National Uniform Claim Committee (NUCC) Health Care Provider taxonomy database. Taxonomy codes were grouped into specialty categories based on the primary 'specialization' and 'classification' fields (Supplementary Data 1). We excluded clinicians with a missing NPI number or primary taxonomy code.

### Statistics and reproducibility
Demographics, Charlson Comorbidity index (as a marker of overall chronic illness burden), frequencies of specific comorbidities relevant to glucocorticoid toxicity, and healthcare utilization metrics were reported using descriptive statistics (Table 1). T-tests were used to compare means for continuous variables, and chi-square was used to compare proportions for categorical variables. We calculated the proportion of a) glucocorticoid users, b) prolonged glucocorticoid users, c) glucocorticoid prescribers at each VA facility over the study period, then stratified facilities by complexity designation, geographic region, and rurality as described above (Figs. 1 and 2, Supplementary Table 2). After attributing filled glucocorticoid prescriptions to prescribing clinicians, we calculated the proportion of prescriptions attributable to each specialty (Fig. 3, Supplementary Data 2). For the three highest-prescribing specialties, we looked at the proportion of glucocorticoids prescribed by each specialty stratified by site complexity (Table 2).

All analyses were performed using R version 4.4.0 (https://github.com/rlipsonVA/PRIDE).

### Reporting summary
Further information on research design is available in the Nature Portfolio Reporting Summary linked to this article.

## Results
### Cohort characteristics
We identified 5,729,134 Veterans who received care at VHA during 2021–2022 and had an assigned VHA PCP. We excluded 75,323 Veterans who received primary care at a facility transitioning to Cerner during the study period, and 292,514 Veterans with no pharmacy claims during the study period. The remaining cohort of 5,361,297 patients was analyzed. Of this cohort, 642,831 patients (12.0%) filled ≥1 oral glucocorticoid prescription during 2021–2022, and 105,846 patients (16% of glucocorticoid

**Table 1 | Baseline characteristics of the analytic cohort stratified by glucocorticoid use**

| | Overall cohort N = 5,361,297 | All GC users* N = 642,831 | Prolonged GC users† N = 105,846 | Non-users N = 4,718,466 |
|---|---|---|---|---|
| Age, mean (SD) | 61 (17) | 61 (15) | 67 (13) | 61 (17) |
| Male sex, N(%) | 4,801,229 (90%) | 560,293 (87%) | 96,313 (91%) | 4,240,936 (90%) |
| White race, N(%) | 3,729,882 (70%) | 439,225 (68%) | 75,688 (72%) | 3,290,657 (70%) |
| GC prescriptions per year, mean (SD) | 1.15 (1.50) | 1.15 (1.50) | 2.93 (2.62) | n/a |
| Charlson comorbidity index, median (IQR) | 0 (0, 1) | 1 (0, 2) | 2 (1, 3) | 0 (0, 1) |
| Outpatient visits per year, median (IQR) | 21 (12, 36) | 36 (22, 56) | 42 (26, 65) | 19 (11, 33) |
| Comorbidities, N(%)[a] | | | | |
| Acute Bronchitis | 101,141 (1.9%) | 52,100 (8.1%) | 4,929 (4.7%) | 49,041 (1.0%) |
| Arterial or venous thrombosis | 154,452 (2.9%) | 34,728 (5.4%) | 9,724 (9.2%) | 119,724 (2.5%) |
| Bacterial infections | 200,639 (3.7%) | 58,130 (9.0%) | 14,577 (14%) | 142,509 (3.0%) |
| Asthma | 293,894 (5.5%) | 73,176 (11%) | 10,337 (9.8%) | 220,718 (4.7%) |
| Cataracts and glaucoma | 1,496,974 (28%) | 231,962 (36%) | 43,012 (41%) | 1,265,012 (27%) |
| Cerebrovascular disease | 295,707 (5.5%) | 47,470 (7.4%) | 9,862 (9.3%) | 248,237 (5.3%) |
| COPD | 675,116 (13%) | 167,352 (26%) | 33,455 (32%) | 507,764 (11%) |
| COVID-19 | 429,118 (8.0%) | 115,664 (18%) | 16,701 (16%) | 313,454 (6.6%) |
| Diabetes | 1,553,518 (29%) | 193,789 (30%) | 39,063 (37%) | 1,359,729 (29%) |
| Fractures | 160,412 (3.0%) | 35,813 (5.6%) | 7,295 (6.9%) | 124,599 (2.6%) |
| Fungal infections | 595,101 (11%) | 107,744 (17%) | 20,374 (19%) | 487,357 (10%) |
| Gout | 315,749 (5.9%) | 74,238 (12%) | 13,197 (12%) | 241,511 (5.1%) |
| Immune-mediated conditions | 795,943 (15%) | 182,673 (28%) | 42,558 (40%) | 613,270 (13%) |
| Inflammatory skin conditions | 522,414 (9.7%) | 106,231 (17%) | 19,214 (18%) | 416,183 (8.8%) |
| Influenza | 22,011 (0.4%) | 8,579 (1.3%) | 959 (0.9%) | 13,432 (0.3%) |
| Malignancy | 814,878 (15%) | 132,390 (21%) | 18,887 (18%) | 682,488 (14%) |
| Osteoporosis | 87,014 (1.6%) | 20,055 (3.1%) | 8,286 (7.8%) | 66,959 (1.4%) |
| Pneumonia | 152,423 (2.8%) | 60,392 (9.4%) | 13,252 (13%) | 92,031 (2.0%) |
| Sepsis | 69,566 (1.3%) | 19,798 (3.1%) | 6,287 (5.9%) | 49,768 (1.1%) |
| Viral infections | 375,314 (7.0%) | 84,320 (13%) | 13,720 (13%) | 290,994 (6.2%) |

*GC* glucocorticoid, *COPD* chronic obstructive pulmonary disease, *COVID-19* coronavirus disease 2019.

*All *p* vs non-use <0.001; †all with p vs non-use <0.001.

[a]Classified using Clinical Classifications Software Refined (CCSR) categories based on the International Classification of Diseases, 10th revision (ICD-10).

users, 2.0% of the analytic cohort) met criteria for prolonged glucocorticoid use. A distribution of treatment duration divided into deciles for the analytic cohort and for glucocorticoid users is presented in Supplemental Data 3.

Characteristics of the analytic cohort, both overall and stratified by glucocorticoid use, are presented in Table 1. This cohort is 90% male and 70% white, with mean (SD) age 61 (17) years, a mean (SD) of 1.15 (1.50) glucocorticoid prescriptions per year, and a median (IQR) of 21 (12, 36) outpatient visits per year.

Compared to non-users, patients who received glucocorticoids during the study period are observed to have more comorbidities (median[IQR] Charlson comorbidity index 1[0,2] vs 0[0,1]) and outpatient visits per year (median[IQR] 36[22,56] vs 19[11,33] visits). Compared to non-users, glucocorticoid users are also observed to have experienced higher rates of bacterial infections (9.0% vs. 3.0%), viral infections (13% vs. 6.2%), cataracts and glaucoma (36% vs. 27%), cerebrovascular disease (7.4% vs. 5.3%), fractures (5.6% vs. 2.6%), influenza (1.3% vs. 0.3%), COVID-19 (18% vs. 6.6%), osteoporosis (3.1% vs. 1.4%), pneumonia (9.4% vs. 2.0%) and sepsis (3.1% vs. 1.1%). Compared to non-users, prolonged GC users are older (mean[SD] age 67(13) vs 61 (17) years). Trends observed for prolonged users are otherwise similar to those for glucocorticoid users overall.

**Patient glucocorticoid use**

Nationally, 1,147,260 glucocorticoid prescriptions were filled in VHA during 2021–2022. The mean (SD) percentage of patients filling ≥1

prescription at each facility is 12.1% (3.5%). Mean (SD) rates stratified by facility complexity are 11.1% (2.7%) at level 1a facilities, 12.6% (3.0%) at level 1b, 14.5% (3.9%) at level 1c, 11.9% (4.7%) at level 2, and 11.1% (2.5%) at level 3 (Fig. 1A). Prescribing rates range between 6.4% and 16.8% at level 1a, 8.9% and 18.9% at level 1b, 8.6 and 26.7% at level 1c, 5.2% and 26.8% at level 2, and 6.3% and 15.5% at level 3. Regional mean (SD) rates of glucocorticoid use range from 9.7% (3.1%) in the Pacific to 13% (2.2%) in the Southeast (Supplementary Table 2). The mean (SD) rate of use among urban facilities is 12.1% (3.5%), and among rural facilities is 12.7% (3.6%).

The mean (SD) percentage of patients meeting criteria for prolonged glucocorticoid use during the study period is 2.1% (0.5%). Mean (SD) rates of prolonged use stratified by facility complexity are 1.9% (0.4) at level 1a facilities, 2.2% (0.5) at level 1b, 2.1% (0.5) at level 1c, 2.1% (0.5) at level 2, and 2.1% (0.5) at level 3 (Fig. 1B). Regional mean (SD) rates of prolonged use range from 1.8% (0.3) in the Southeast to 2.4% (0.4) in the Midwest (Supplementary Table 2). The mean (SD) rate of prolonged use among urban facilities is 2.0% (0.4) and among rural facilities is 2.3% (0.6).

**Provider glucocorticoid prescribing**

Between 2021–2022, 400,135 clinicians wrote ≥1 prescription filled by a patient in the analytic cohort above. Of these, 82,071 wrote ≥1 glucocorticoid prescription, and 65,448 (84.2% of glucocorticoid prescribers) had an NPI number linked to a NUCC taxonomy code and were analyzed. Across facilities, the mean (SD) percentage of clinicians who prescribed

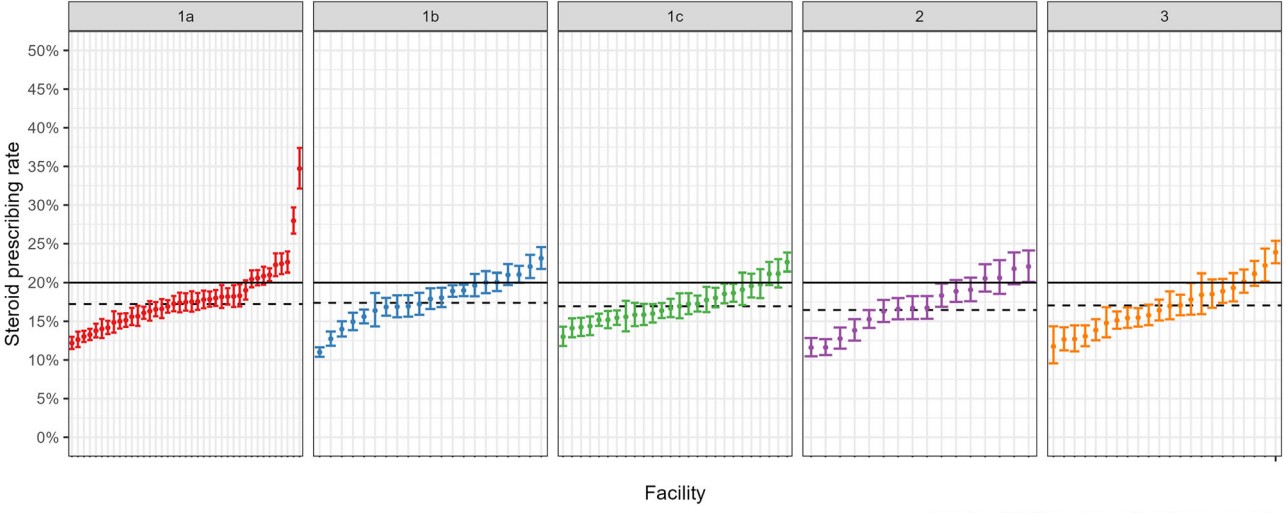

**Fig. 1 | Facility-level rates of glucocorticoid use stratified by facility complexity. A** shows overall glucocorticoid use. **B** shows prolonged glucocorticoid use. The solid lines represent the overall mean. The dotted lines represent the means within each complexity designation. $N = 124$ facilities. Facility complexity ranges from 1a to 3.

Excluding 34,294 providers with unknown specialty

**Fig. 2 | Facility-level rates of glucocorticoid prescribing, stratified by facility complexity.** The solid lines represent the overall mean. The dotted lines represent the means within each complexity designation. $N = 124$ facilities. Facility complexity ranges from 1a to 3.

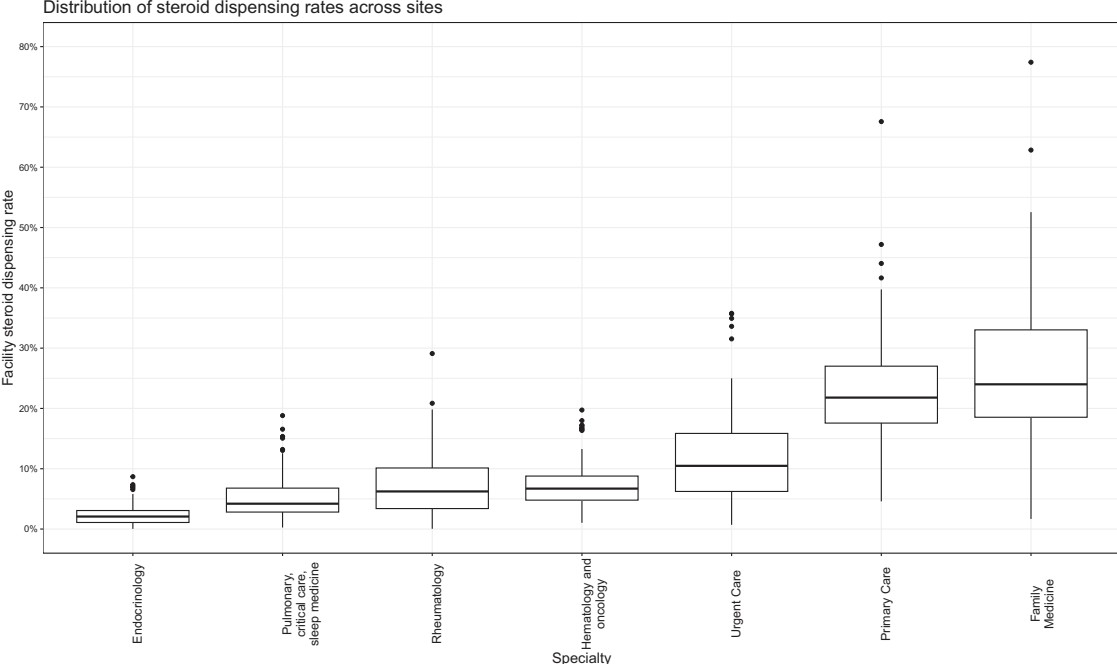

**Fig. 3 | Prescribing rates for the specialties prescribing the most glucocorticoids across sites.** Each box represents the interquartile range of steroid dispensing rate for a given specialty. The solid line inside the box represents the median rate for that specialty. The whiskers above and below the box represent the upper and lower quartile rates, respectively. The solid circles above and below the whiskers represent outliers.

**Table 2 | Percentage of total glucocorticoids prescribed by each of the three highest-prescribing specialties, by site complexity**

| Site complexity | Mean (SD) percentage of total glucocorticoids prescribed during the study period | | |
|---|---|---|---|
| | **Family medicine** | **Urgent care** | **Primary care** |
| 1a | 20.7 (8.1) | 11.3 (5.9) | 24.7 (5.9) |
| 1b | 21.6 (8.4) | 13.8 (7.2) | 23.1 (5.8) |
| 1c | 25.7 (10.7) | 15.6 (8.5) | 21.7 (8.3) |
| 2 | 30.9 (11.9) | 9.6 (8.5) | 19.4 (9.0) |
| 3 | 38.0 (15.3) | 6.2 (5.7) | 23.5 (14.4) |

glucocorticoids during 2021–2022 was 17.4% (3.4). Mean (SD) rates of prescribing stratified by facility complexity are 17.9% (4.2) at level 1a facilities, 17.8% (3.1) at level 1b, 17.1% (2.4) at level 1c, 17.0% (3.4) at level 2, and 16.9% (3.2) at level 3 (Fig. 2). Prescribing rates range between 12.2.% and 34.7% at level 1a, 11.0% and 23.7% at level 1b, 13.0% and 22.6% at level 1c, 11.6% and 22.0% at level 2, and 11.7% and 23.9% at level 3. Regional mean (SD) rates of prescribing range from 15.2% (2.6) in the North Atlantic, to 19.6% (4.3) in the Southeast. (Supplementary Table 2) The mean among urban facilities is 17.4% (3.5), and among rural facilities is 17.5% (2.6).

Family medicine (n = 12,302 clinicians, 18.8% of the clinician cohort) was responsible for 26.3% of all glucocorticoid prescriptions filled during 2021–2022. Non-family medicine PCPs (n = 14,045, 21.5%) were responsible for 22.9% of all glucocorticoid prescriptions. Urgent care (n = 6459, 9.9%) were responsible for 11.6% of prescriptions, rheumatology (n = 1446, 2.2%) for 7.4%, hematology/oncology (n = 3606, 5.5%) for 7.3%, pulmonary and critical care (n = 2553, 3.9%) for 5.1%, and endocrinology (n = 1084, 1.6%) for 2.2%. (Fig. 3) Other specialties were each responsible for <2% of glucocorticoid prescriptions. (Supplementary Data 2).

When stratified by facility complexity designation, the mean (SD) percentage of glucocorticoids prescribed by family medicine ranges from 20.7% (8.1%) at high-complexity level 1a facilities to 38.0% (15.3%) at low-

complexity level 3 facilities. Mean (SD) glucocorticoid prescribing rates for other PCPs range from 19.4% (9.0%) at low-complexity level 2 facilities compared to 24.7% (5.9%) at high-complexity level 1a facilities. Mean (SD) glucocorticoid prescribing rates for urgent care range from 6.2% (5.7%) at low-complexity level 3 facilities to 13.8% (7.2%) at higher-complexity level 1b facilities. (Table 2).

## Discussion

In a national population of Veterans, approximately one out of every eight patients filled a glucocorticoid prescription between 2021 and 2022. Of these, one in six meet criteria for prolonged glucocorticoid use, defined as at least 30 days of use in a 365-day period. Compared to non-users, glucocorticoid users are observed to have a substantially higher prevalence of pneumonia (5 times higher), influenza (4 times higher), bacterial infections (3 times higher), fractures, osteoporosis, sepsis, and COVID-19 (2 times higher), cerebrovascular disease (1.3 times higher), and cataracts and glaucoma (1.3 times higher). Rates of glucocorticoid use and prescribing vary substantially both within and across facility complexity designations, with less variation seen across geographic region and level of rurality. Family medicine and other PCPs are together responsible for 49.2% of all glucocorticoid prescriptions during the study period, with urgent care responsible for an additional 11.6%. Family medicine providers wrote 26.3% of filled glucocorticoid prescriptions but represented only 18.8% of glucocorticoid prescribers.

We found a mean glucocorticoid prescribing rate of 12.1% between 2021–2022, compared to annual rates of 6.4%–7.7% reported in commercial claims between 2009–2018, and an overall rate of 1.2% by NHANES participants between 1999–2008[4,14,15]. While prescribing increased steadily over this period, these trends suggest the rate of increase may be stabilizing. Potential reasons include advancing awareness of the harms caused by short-term and low-dose glucocorticoid use[1,5], the discovery of an association between glucocorticoid use and increased risk of COVID-19 related hospitalization[22], and the ongoing development of glucocorticoid-sparing treatments, both for common conditions like asthma, COPD, and psoriasis and rare conditions like multiple sclerosis, lupus, and other autoimmune diseases[23–27].

Over 60% of glucocorticoids dispensed during the study period were prescribed by non-specialists; family medicine, other PCPs, or urgent care. These rates are comparable to those previously seen in the civilian sector, suggesting the novel facility-level trends in our analysis may be generalizable to insured civilians[14]. We found substantial variation in rates of glucocorticoid use both within and across facilities with varying complexity designations, with the highest prescribing at level 1c facilities and comparable rates across levels 1a, 2, and 3. One explanation for this may be that level 2 and 3 facilities refer complex patients to level 1a for specialty care, while level 1b and 1c facilities often manage these patients themselves despite lower volume and reduced provider connectedness to a network of high-prescribing clinical experts. This is in contrast to little variation between geographic regions or urban vs rural facilities, in context of VHA's robust telehealth and virtual consultation expansion over the past decade targeted to address regional and rural-urban disparities in care[28,29].

As expected, we see higher rates of overall glucocorticoid use among a) patients with chronic conditions treated with glucocorticoids when exacerbated (e.g., asthma), and b) chronic conditions managed long-term with glucocorticoids (e.g., malignancy). However, prolonged glucocorticoid use is also higher in these populations. This suggests that long-term glucocorticoid use to manage conditions like gout, COPD and asthma remains common, despite highly effective and increasingly available glucocorticoid-sparing treatment options. While the high cost of glucocorticoid-sparing treatments may provide an obstacle to widespread adoption in the community, in the VHA individual cost of these medications is less often a barrier to access[30]. Clinicians at low-complexity facilities with few specialty pharmacists or specialty care networks may also be less aware or comfortable with glucocorticoid-sparing treatments[31–33].

Despite both providing primary care to patients, family practitioners and non-family practice PCPs demonstrate different prescribing trends by facility complexity designation. Glucocorticoid prescribing by family practice is higher in lower-complexity facilities, suggesting these clinicians may manage certain conditions like COPD or gout independently at these facilities. In contrast, specialists are more readily accessible at high complexity sites, and family practitioners at these sites may be more likely draw on stronger connections, guidance, and high-value glucocorticoid sparing practices from specialty networks to moderate low-value glucocorticoid prescribing. Variation within a facility complexity designation is also highest at facilities with the lowest complexity designation, reinforcing the idea that weaker connections to specialty and high-prescribing clinicians allows for more variability in prescribing practice[16,32,34]. Further, PCPs who were not family practitioners did not vary in glucocorticoid prescribing across facility complexity designation. This may be because these PCPs have more ready access to robust specialist referral networks, and/or more experience or comfort prescribing glucocorticoid-sparing therapies[33]. Urgent care clinicians display similar trends to family practice at high complexity sites, but a reversed trend at lower complexity sites. This may be because patients with chronic complex conditions preferentially go to high-complexity facilities for urgent care, even if their PCP is located at a less complex (perhaps more convenient) facility[35].

Our findings highlight opportunities to reduce avoidable glucocorticoid prescribing. Established opioid and antibiotic stewardship models have led to effective interventions limiting overprescribing[36–38]. Applying these concepts to glucocorticoid stewardship has the potential to prevent avoidable harm by reducing low-value glucocorticoid prescribing. Examples of interventions well-suited to glucocorticoid stewardship initiatives include educational programs or decision aids targeting high-prescribing providers and common indications for overprescribing, practice goals set benchmarked against previous prescribing patterns, clinician-peer comparisons stratified by practice setting and prescribing indication, and third-party prescription monitoring programs incorporating relevant stakeholders. Hub-and-spoke models of care to strengthen connections between centrally located specialists and local primary care providers caring for patients in rural and underserved areas could also help to standardize clinical practices such as glucocorticoid prescribing[39].

This study has many strengths. To our knowledge, this is the largest study examining glucocorticoid use and prescribing patterns in the US. We used claims data from an integrated health system (VHA), allowing us to capture pharmacy dispensing rather than just prescription claims, to link individual clinicians to filled prescriptions reliably, and to examine use patterns based on an established facility-level measure of case-mix and complexity of care. The population served by VHA also includes demographics under-represented in commercial claims data but over-represented among glucocorticoid users, such as elderly and highly comorbid patients. Limitations of this study are primarily related to the use of claims data, including a lack of information about how patients take glucocorticoids once their prescription is filled, limited granular data about the indication for prescribing, an inability to determine a causal relationship between glucocorticoid exposure and rates of conditions associated with glucocorticoid toxicity (e.g., cardiovascular disease, diabetes), potential for misclassification of provider specialty, inability to evaluate providers without an identifiable specialty, and inability to differentiate training type within specialty (e.g., physician, nurse practitioner, physician assistant). We also did not examine glucocorticoid prescriptions filled by Veterans outside the VA pharmacy system, e.g., in the civilian sector or through mechanisms such as fee-basis care, though prior work suggests this is relatively rare[40].

In a national population of over 5 million Veterans, one out of every eight patients filled a glucocorticoid prescription during 2021–2022, and one in fifty met the criteria for prolonged glucocorticoid use. Compared to non-users, glucocorticoid users are at substantially higher risk of comorbidities known to be associated with glucocorticoid exposure. Over a quarter of prescribing clinicians are family practice physicians, who display higher prescribing when practicing at lower-complexity sites where specialty care is not available and relational networks with higher-volume clinicians may be weaker. Educational interventions to improve knowledge of novel glucocorticoid-sparing treatments and/or access to high-expertise peers may reduce glucocorticoid overuse among this subset of clinicians. In addition, work to understand provider preferences around glucocorticoid use may be used to develop interventions that reduce low-value glucocorticoid prescribing and associated avoidable toxicity.

## Data availability

This publication uses, in compliance with local and federal regulatory and legal frameworks, administrative data derived from the medical records of Veterans receiving care at federally funded Veterans Health Affairs medical centers. The Department of Veterans Affairs prevents dataset disclosures to other entities without (1) a data transfer agreement, (2) deidentification of the data set, and (3) appropriate institutional review board approvals. Investigators wishing to obtain these data should contact the corresponding author to discuss the request. The numerical results underlying the graphs and charts presented in the main figures (source data) are made available as Supplementary Data 4–7.

## Code availability

Code for the development of this paper can be found at https://github.com/rlipsonVA/PRIDE.

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

## Acknowledgements

Dr. Wallace's effort during the time of this work was supported by VA ORD CX002430.

## Author contributions

B.W. contributed to study conception and design, data analysis and interpretation, manuscript drafting. S.C.M. contributed to data interpretation, critical revision of manuscript, final version approval. T.V., R.L. and B.K. contributed to data analysis and interpretation, critical revision of manuscript, final version approval. C.T. and J.B. contributed to critical revision of manuscript, final version approval. S.S. contributed to data acquisition, critical revision of manuscript, final version approval. A.K.W. contributed to study conception and design, data acquisition, critical revision of manuscript, final version approval.

## Competing interests

The authors declare no competing interests.

## Additional information

[1]Center for Clinical Management Research, VA Ann Arbor Health Care System, Ann Arbor, MI, USA. [2]Department of Internal Medicine, VA Ann Arbor Health Care System, Ann Arbor, MI, USA. [3]Department of Internal Medicine, University of Michigan, Ann Arbor, MI, USA. [4]Department of Survery, University of Michigan, Ann Arbor, MI, USA. [5]Department of Learning Health Sciences, University of Michigan Medical School, Ann Arbor, MI, USA. ✉e-mail: brennerb@umich.edu

