## [Peer Review file · Communications Medicine]

The role of clinician specialty and facility complexity on glucocorticoid prescribing: a national cross-sectional study of Veterans during 2021-2022

Corresponding Author: Dr Beth Wallace

Version 0:

Reviewer comments:

Reviewer #1

(Remarks to the Author)

Bold Start Major Issues and Minor Comment

This study utilized Veterans Health Administration (VHA) data to quantify glucocorticoid (GC) use and explore the impact of facility complexity and clinician specialty on oral glucocorticoid prescribing patterns in the VA population. While the study design and dataset are robust, several key issues should be addressed to maximize the utility of the extensive data available.

1. Distribution of Glucocorticoid Use: Describing prolonged GC users is informative, but readers, particularly clinicians, would benefit from a more detailed analysis of GC usage across the entire cohort. A distribution of GC treatment duration divided into quantiles or increments of 10% would provide a comprehensive view of treatment patterns. This approach would allow for meaningful comparisons with other patient groups or regions and give a clearer picture of the variability in GC use.
2. Adverse Drug Events and GC Types: While the authors correctly highlight the risks associated with GC use, it is unclear if all GC drugs carry the same rate and severity of adverse events. The study does not provide details on the types and routes of administration of the various GCs in the dataset. Reporting this information, including the most commonly prescribed GCs at the VHA, would be valuable to readers. Additionally, the authors should explore whether prescribing patterns vary by specialty or across different regions of the VA system.
3. Prescription Indices for GC Drugs: The authors have the opportunity to enhance their analysis by including prescription indices such as the Prescription Selectivity Index and the Herfindahl Index. These indices could better quantify prescribing patterns, offering insights into the concentration and variability of GC prescribing across specialties, patient age groups, and geographic regions. Such analyses would add depth to the findings and provide useful metrics for comparing different populations and subpopulations.
4. Methodology Clarification: In the methods section, there is a sentence that states, "chi-square was used to compare means for categorical variables." This is inaccurate, as chi-square tests are typically used to compare proportions or distributions of categorical variables, not means. Please revise this sentence.
5. Statistical Testing and Significance Results: The authors report statistically significant results in Table 1. Given the large sample size, statistical significance may be achieved even for small differences. It would be more meaningful to focus on comparisons with sizable differences. Please review Table 1, as it appears all comparisons are statistically significant, which could be a result of the sample size rather than meaningful differences.
6. Caution on Causal Interpretation in the Discussion: In the discussion section, the results should be interpreted with caution, particularly to avoid implying causality. For example, the statement that "GC users had more bacterial infections, fractures, osteoporosis" compared to non-users could suggest that GC use caused these conditions. However, since this is a cross-sectional study, it is impossible to determine whether GCs led to these outcomes or if GCs were prescribed for pre-existing conditions (reverse causality).
7. Minor Comment: There is a typo in the definition of eTable 1. The VA class code should be HS051 (glucocorticoids), not HS501 (insulin).

Reviewer #2

(Remarks to the Author)

In this study the authors sought to describe the effect of facility complexity and clinician specialty on oral glucocorticoid steroid prescribing among Veterans who receive care within the VA healthcare system. Among 5,729,134 patients at 124 facilities, a mean of 12.1% (SD 3.5%) were glucocorticoid users and 2.1% (SD 0.5%) were prolonged users. Rates of use

and prescribing varied substantially within and across facility complexity designations. Family practice represented 18.8% of glucocorticoid-prescribing clinicians but prescribed 26.3% of filled prescriptions. Their findings support the conclusion that clinician-facing interventions to address knowledge gaps, incentivize non-glucocorticoid treatments, or facilitate specialty care access may reduce overuse among high-prescribing clinicians.

Overall, this is a well-written study that is relatively straightforward in its design. It effectively leverages the rich data available for the millions of Veterans that VA services. It is motivated by the potential harms associated with glucocorticoid overprescribing and the relatively novel concept of glucocorticoid stewardship.

While I feel that overall, the study is well done, the authors should consider the following comments to further strengthen the manuscript.

1)The authors should further justify or revisit the decision to include only those Veterans who received at least 1 prescription from VA during the study period. This resulted in the exclusion of nearly 300,000 Veterans who, because they still received primary care in VA, would still have the opportunity to receive a steroid should it be indicated.

2)The study was described as a cohort study, but I felt the design was more cross-sectional in nature, as we are not really studying how the receipt of steroids affect Veterans' health outcomes.

3)I recommend the authors describe their data sources in somewhat more detail. They allude to pulling their claims from the corporate data warehouse, but there are other files, such as the fee basis file or program integrity tool that may be used to capture claims paid for by VA but delivered by non VA-sources. Were any of these files contained in their analyses?

4)The authors need to clarify the definition of prolonged glucocorticoid use. In the discussion, they indicate that prolonged use requires 30 days of consecutive use, but this was not mentioned in the methods. However, the decision to only include those with 30 days of consecutive use should be further justified. For example, if this is the case, this could exclude a patient with persistent asthma who has received multiple steroid tapers throughout the year. This is likely a person who would be at increased risk of adverse outcomes related to steroid use but wouldn't meet prolonged use criteria as presently defined.

5)I recommend including a sentence in the first paragraph of the results describing the demographics/characteristics of the overall cohort. This will put the description of glucocorticoid users in greater context.

6)The authors make reference to the Charlson comorbidity index in the results, but this is not mentioned in the methods.

7)Suggested revisions to the Discussion:

a. In the discussion, the authors do a good job generating hypotheses and explanations for findings. However, I would have liked more discussion about the implications of the study findings. Putting myself in the shoes of a VA clinical leader/operations partner, I found myself questioning what I should do with these results. For example, how will these findings potentially impact clinical care within VA. Does this support the development/further dissemination of a glucocorticoid stewardship program? If so, how? What are the implications for care outside VA?

b. Relatedly, the authors identified that family medicine physicians tend to prescribe more steroids than other specialties. But how functionally distinct are family physicians from other primary care specialties within VA? For example, our primary care service line contains internal medicine and family medicine trained physicians who function similarly. I raise this because this has implications for how the study findings may be used in practice.

c. Also, did the authors consider looking at differences in prescribing patterns among MDs/DOs vs advanced practice providers? This could yield some interesting patterns and could inform dissemination of the findings.

d. In the discussion, the authors report the study supports that steroid use may be stabilizing, but I do not feel the data support this, as the prescribing rate of steroids they report is nearly 50% higher than the rate reported in claims in the prior decade.

8)In Figures 1 and 2 the numbering of the X axis are too small to read. If the authors are striving to demonstrating variability across VA Medical Centers, I don't think they need to include these specific facility numbers in the X axis – the plot speaks for itself.

Reviewer #3

(Remarks to the Author)

The purpose of the manuscript is to describe the effect of facility complexity and clinician specialty on the prevalence of glucocorticoid use and patterns in a national population of US Veterans between 2021 and 2022. This is an important study as the authors note for several reasons including population level adverse health risk increases.

This study provides a good descriptive accounting on glucocorticoid use using stratified analysis. I would recommend applying a regression model to quantify the contribution of the multiple variables and the relationships between the variables to strengthen the study.

Overall, this study provides evidence to the impact glucocorticoid use has on the population and may help provide support for "glucocorticoid stewardship" as the authors note model similar to antibiotic and opiate prescribing.

Version 1:

Reviewer comments:

Reviewer #1

(Remarks to the Author)

All my comments and suggestions were taken into consideration in the authors' response letter and the changes were made on revised manuscript and supplemental materials.

Reviewer #2

(Remarks to the Author)

This is a well-written study that effectively leverages the rich data available for the millions of Veterans that VA services. It is motivated by the potential harms associated with glucocorticoid overprescribing and the relatively novel concept of glucocorticoid stewardship. The authors have been thoughtful in their responses to the prior reviewer comments. They have further justified and clarified a number of points and have enhanced the discussion by more thoroughly exploring the implications of their work.

Reviewer #3

(Remarks to the Author)

I am satisfied with the responses to all reviewers comments.

Reviewer #1

Q1. Distribution of Glucocorticoid Use: Describing prolonged GC users is informative, but readers, particularly clinicians, would benefit from a more detailed analysis of GC usage across the entire cohort. A distribution of GC treatment duration divided into quantiles or increments of 10% would provide a comprehensive view of treatment patterns. This approach would allow for meaningful comparisons with other patient groups or regions and give a clearer picture of the variability in GC use.

R1: We have generated distributions of glucocorticoid treatment duration across a) the entire analytic cohort, b) glucocorticoid users, divided into deciles. In the larger cohort, 80% of patients received 0 days of glucocorticoids during the study period, and the 90th percentile received 1 to 5 days' supply. Among glucocorticoid users, the bottom 50% of patients have glucocorticoid treatment durations of one week or less, and the bottom 70% have glucocorticoid treatment durations of 15 days or less. We have also added this information to our supplementary data as eTable 5, referenced on page 7 of our Results section.

Table I: Deciles of GC treatment duration

Decile	Duration of receipt by patient (days)	
	Analytic Cohort N = 5,729,134	Glucocorticoid users N = 642,831
1	0	1 to <5
2	0	5
3	0	5 to <6
4	0	6
5	0	6 to <7
6	0	7 to <11
7	0	11 to <15
8	0	15 to <30
9	1 to 5	30 to <120
10	6 to 2640	120 to 2640

Q2. Adverse Drug Events and GC Types: While the authors correctly highlight the risks associated with GC use, it is unclear if all GC drugs carry the same rate and severity of adverse events. The study does not provide details on the types and routes of administration of the various GCs in the dataset. Reporting this information, including the most commonly prescribed GCs at the VHA, would be valuable to readers. Additionally, the authors should explore whether prescribing patterns vary by specialty or across different regions of the VA system.

R2: VA claims data reliably captures outpatient glucocorticoid prescriptions, which include oral, inhaled, and topical (otic, ophthalmic, cutaneous, nasal, etc.) formulations. Given limited data on the systemic absorption of inhaled and topical glucocorticoids, we limited our analyses of outpatient glucocorticoid prescriptions to oral glucocorticoids. We did not include glucocorticoids administered at healthcare centers, e.g. intravenous and intramuscular glucocorticoids, in our analyses as their administration is difficult to accurately determine using claims data. We have clarified this in the introduction and methods, with changes including addition of the following sentence to our methods section (page 5):

“We limited our evaluation to oral glucocorticoids given limited data on the systemic absorption of inhaled and topical glucocorticoids, and limitations of claims data for characterizing exposure to intravenous and intramuscular glucocorticoids.”

Oral glucocorticoids are therapeutically interchangeable on the basis of all attributes but potency and half-life. They have been commercially available since the 1940s, and all glucocorticoid drugs are available in VA, and primarily used, in generic form. Prednisone is by far the most commonly prescribed oral glucocorticoid

formulation in the VA (and the United States more generally) as it has intermediate potency and is dosed once daily. Other oral glucocorticoids are typically reserved for specific clinical scenarios, e.g. dexamethasone is used for indications requiring a high-potency oral steroid (e.g. severe allergic reactions, oncologic indications), hydrocortisone is used for management of adrenal insufficiency because its half-life mimics physiologic secretion etc. As a rule, clinicians do not consider market share or other branding concerns when prescribing oral glucocorticoids and their side effect profiles are regarded as comparable.

The table below shows the frequency of the oral glucocorticoid formulations filled by our study population. Eighty-seven percent of the prescriptions filled were for prednisone, 8% were for dexamethasone, and 3% were for hydrocortisone. Remaining formulations accounted for less than 2% of total prescriptions dispensed.

Table II: Frequency of oral glucocorticoid formulations prescribed

Formulation	Prescriptions during study period (N, %)
Prednisone	1,288,208 (87.34%)
Dexamethasone	117,249 (7.95%)
Hydrocortisone	44,621 (3.01%)
Methylprednisolone	23,291 (1.58%)
Prednisolone	434 (0.029%)
Triamcinolone	308 (0.021%)

Q3. Prescription Indices for GC Drugs: The authors have the opportunity to enhance their analysis by including prescription indices such as the Prescription Selectivity Index and the Herfindahl Index. These indices could better quantify prescribing patterns, offering insights into the concentration and variability of GC prescribing across specialties, patient age groups, and geographic regions. Such analyses would add depth to the findings and provide useful metrics for comparing different populations and subpopulations.

R3: While the Prescription Selective Index and the Herfindahl Index are both helpful in comparing drug toxicity-to-effect and comparing brand formulations, respectively, their role in furthering our study objectives are limited.

The Prescription Selectivity Index is a ratio that measures the relative toxicity of a substance compared to its intended effect (e.g. cytotoxicity for chemotherapy, antiviral activity for antibiotics). It was developed to enable comparisons between drugs with the same therapeutic purpose but different mechanisms of action and/or toxicity profiles. Glucocorticoid steroids form a single drug class with a common mechanism of action and a common toxicity profile. Additionally, glucocorticoids are used for a wide variety of clinical indications, making it difficult to assess “effectiveness”. For example, a glucocorticoid prescription for rheumatoid arthritis is “effective” if it reduces joint swelling, while a prescription for cancer anorexia is “effective” if it increases appetite.

The Herfindahl index is used to compare market concentration/competition within a specific drug category. Glucocorticoids have been commercially available since the 1940s, and all drugs in the class are available, and primarily used, in generic form. While it is possible to prescribe brand-name glucocorticoid formulations with specific niche features (e.g. delayed release), these are generally not available on the VA formulary and not commonly used in the United States.

Q4. Methodology Clarification: In the methods section, there is a sentence that states, "chi-square was used to compare means for categorical variables." This is inaccurate, as chi-square tests are typically used to compare proportions or distributions of categorical variables, not means. Please revise this sentence.

R4: Thank you for pointing out this error. We have changed this statement as follows (page 6):

“...chi-square was used to compare proportions for categorical variables”.

Q5. Statistical Testing and Significance Results: The authors report statistically significant results in Table 1. Given the large sample size, statistical significance may be achieved even for small differences. It would be more meaningful to focus on comparisons with sizable differences. Please review Table 1, as it appears all comparisons are statistically significant, which could be a result of the sample size rather than meaningful differences.

R5: We agree that our large sample size makes statistical comparisons less useful than comparisons of effect size. To allow the reader to focus on the effect size of the differences between the groups, we removed language reporting statistical significance from the Results section of the manuscript (page 7) and removed the footnote stating that all comparisons were significant from Table 1.

Q6. Caution on Causal Interpretation in the Discussion: In the discussion section, the results should be interpreted with caution, particularly to avoid implying causality. For example, the statement that “GC users had more bacterial infections, fractures, osteoporosis” compared to non-users could suggest that GC use caused these conditions. However, since this is a cross-sectional study, it is impossible to determine whether GCs led to these outcomes or if GCs were prescribed for pre-existing conditions (reverse causality).

R6: We agree with the Reviewer’s cautions about interpreting our observations causally, given the design of the study. The verbiage above is taken from our Results section (page 7). We have amended it as follows. We have also removed language around statistical testing from this section, per Reviewer 1’s recommendation.

“Compared to non-users, patients who received glucocorticoids during the study period were observed to have more comorbidities...and outpatient visits. Compared to non-users, glucocorticoid users were also observed to have experienced higher rates of bacterial infections...viral infections...[etc]”

In our discussion section, we describe these trends as follows (page 10)

“Compared to non-users, glucocorticoid users were observed to have a significantly higher prevalence of pneumonia (5 times higher), influenza (4 times higher), bacterial infections (3 times higher), fractures, osteoporosis, sepsis, and COVID-19 (2 times higher), cerebrovascular disease (1.3 times higher), and cataracts and glaucoma (1.3 times higher).”

In our discussion, we also cite as a limitation our inability to make causal inferences from descriptive data (page 12).

“Limitations of this study [include]...an inability to determine a causal relationship between glucocorticoid exposure and rates of conditions associated with glucocorticoid toxicity”

Q7. Minor Comment: There is a typo in the definition of eTable 1. The VA class code should be HS051 (glucocorticoids), not HS501 (insulin).

R7: Thank you for pointing this out. We have corrected the typo in eTable 1.

Reviewer #2

Overall, this is a well-written study that is relatively straightforward in its design. It effectively leverages the rich data available for the millions of Veterans that VA services. It is motivated by the potential harms associated with glucocorticoid overprescribing and the relatively novel concept of glucocorticoid stewardship.

Thank you for your constructive feedback!

Q1) The authors should further justify or revisit the decision to include only those Veterans who received at least 1 prescription from VA during the study period. This resulted in the exclusion of nearly 300,000 Veterans who, because they still received primary care in VA, would still have the opportunity to receive a steroid should it be indicated.

R1: The objective of our study was to examine patterns of glucocorticoid use in the Veteran population. The majority of our analyses, with the exception of Table 1, focus on glucocorticoid users. Glucocorticoid users represent 12% of our analytic cohort as defined (642,831 of 5,361,297). By definition, the 292,514 Veterans who did not have a pharmacy claim during the study period are glucocorticoid non-users. If we re-included these Veterans, our denominator would become 5,653,811, a 5% difference. Our glucocorticoid cohort would be the same, and the new prevalence of glucocorticoid use using the expanded denominator would be 11.3%. This difference is quite small. Additionally, the patients in question are likely to a) have little contact with the healthcare system, b) receive care from both VA and non-VA provider(s), or both. As a result, these patients likely do not represent typical Veterans eligible for steroid use within the VA.

For these reasons, we feel analysis of patterns of care among patients who did not receive prescriptions during the study period is outside the scope of this work. We have clarified this by amending our Abstract (page 2) and background section (Page 4) as follows:

Abstract: "This is a national cross-sectional study of Veterans prescribed oral glucocorticoids during 2021-2022."

Background: "Specifically, we build on previous studies by evaluating how clinician specialty and facility complexity designation affect oral glucocorticoid use and prescribing patterns among Veterans who received glucocorticoid treatment."

Q2) The study was described as a cohort study, but I felt the design was more cross-sectional in nature, as we are not really studying how the receipt of steroids affects Veterans' health outcomes.

R2: We agree that it is more appropriate to describe this study as cross-sectional, as it does not evaluate associations between factors examined or look at change over the study period. We have clarified this throughout the manuscript including the title.

Q3) I recommend the authors describe their data sources in somewhat more detail. They allude to pulling their claims from the corporate data warehouse, but there are other files, such as the fee basis file or program integrity tool that may be used to capture claims paid for by VA but delivered by non VA-sources. Were any of these files contained in their analyses?

R3: We used only CDW files for the data in this manuscript. We did not query the fee basis file or any other files capturing non-VA care for Veterans (e.g. Program Integrity Tool, VA linked Medicare data, etc.). We clarified this in our limitations (page 12):

“We also did not examine glucocorticoid prescriptions filled by Veterans outside the VA pharmacy system, e.g. in the civilian sector or through mechanisms such as fee-basis care”

Q4) The authors need to clarify the definition of prolonged glucocorticoid use. In the discussion, they indicate that prolonged use requires 30 days of consecutive use, but this was not mentioned in the methods. However, the decision to only include those with 30 days of consecutive use should be further justified. For example, if this is the case, this could exclude a patient with persistent asthma who has received multiple steroid tapers throughout the year. This is likely a person who would be at increased risk of adverse outcomes related to steroid use but wouldn't meet prolonged use criteria as presently defined.

R4: Thank you for noticing this wording error. Our definition of prolonged glucocorticoid use did not require 30 days of consecutive use. As previously published, it required 30 days of use or more in a 365 day period. (Wallace et al, JAMA Network Open 2021) This definition is met by 30 days of consecutive use but is also met by multiple shorter steroid courses as the Reviewer outlines above. We have corrected the referenced sentence in the discussion as follows (page 10):

“Of these, one in six met criteria for prolonged glucocorticoid use, defined as at least 30 days of use in a 365-day period.”

Q5) I recommend including a sentence in the first paragraph of the results describing the demographics/characteristics of the overall cohort. This will put the description of glucocorticoid users in greater context.

R5: We have added the sentence below to our Results section. We also clarified as below that Table 1 provides demographics and characteristics for the overall cohort (N = 5,361,297) in addition to those stratified by glucocorticoid use (page 7):

“Characteristics of the analytic cohort, both overall and stratified by glucocorticoid use, are presented in Table 1. This cohort was 90% male and 70% white, with mean (SD) age 61 (17) years, a mean (SD) of 1.15 (1.50) glucocorticoid prescriptions per year, and a median (IQR) of 21 (12, 36) outpatient visits per year.”

Q6) The authors make reference to the Charlson comorbidity index in the results, but this is not mentioned in the methods.

R6: Thank you for pointing this out. We have clarified in our Methods section that we report both the frequencies of specific comorbidities relevant to glucocorticoid toxicity, and the Charlson Comorbidity Index in our population as a marker of overall burden of chronic illness (page 6)

“Demographics, Charlson Comorbidity index (as a marker of overall chronic illness burden), frequencies of specific comorbidities relevant to glucocorticoid toxicity, and healthcare utilization metrics were reported using descriptive statistics (Table 1)”

Q7) Suggested revisions to the Discussion:

a. In the discussion, the authors do a good job generating hypotheses and explanations for findings. However, I would have liked more discussion about the implications of the study findings. Putting myself in the shoes of a VA clinical leader/operations partner, I found myself questioning what I should do with these results. For example, how will these findings potentially impact clinical care within VA.

Does this support the development/further dissemination of a glucocorticoid stewardship program? If so, how? What are the implications for care outside VA?

R7a: We have added a paragraph to the Discussion section highlighting the implications of our study findings (page 13):

Our findings highlight opportunities to reduce avoidable glucocorticoid prescribing. Established opioid and antibiotic stewardship models have led to effective interventions limiting overprescribing. Applying these concepts to glucocorticoid stewardship has the potential to prevent avoidable harm by reducing low-value glucocorticoid prescribing. Examples of interventions well-suited to glucocorticoid stewardship initiatives include educational programs or decision aids targeting high-prescribing providers and common indications for overprescribing, practice goals set benchmarked against previous prescribing patterns, clinician-peer comparisons stratified by practice setting and prescribing indication, and third-party prescription monitoring programs incorporating relevant stakeholders. Hub-and-spoke models of care to strengthen connections between centrally located specialists and local primary care providers caring for patients in rural and underserved areas could also help to standardize clinical practices such as glucocorticoid prescribing.

b. Relatedly, the authors identified that family medicine physicians tend to prescribe more steroids than other specialties. But how functionally distinct are family physicians from other primary care specialties within VA? For example, our primary care service line contains internal medicine and family medicine trained physicians who function similarly. I raise this because this has implications for how the study findings may be used in practice.

R7b: We constructed our specialty designation categories using provider NPI cross-referenced with NUCC taxonomy codes. We classified steroid-prescribing providers with “Internal Medicine (N = 12,904)” or “General Practice (N = 374)” taxonomy codes as “Primary care”, and those with “Family Medicine” (N = 5,474) taxonomy codes as “Family Medicine”. We recognize that provider misclassification and missing data are limitations of this methodology, and added a sentence to our discussion section to emphasize this (page 12):

“Limitations of this study [include]... potential for misclassification of provider specialty, and inability to evaluate providers without an identifiable specialty.”

We agree that, at least in VA, internal medicine and family medicine-trained providers are not explicitly differentiated in primary care service lines. However, family medicine and internal medicine providers are trained differently, and published data suggest that family medicine providers tend to be over-represented in certain geographic regions (e.g. the southeast) and practice locations (e.g. rural and resource-constrained settings) relative to internal medicine. (Xierali et al, J Health Care Poor Underserved, 2018) We confirmed these trends in our own data, as shown below. It is thus reasonable to assume that even if family medicine and internal medicine providers are not filling explicitly different roles at an administrative level, differences in practice style may develop as a function of other factors, e.g. training, regional variation, and overrepresentation of family medicine providers in rural and underserved areas.

Table IV: Providers prescribing glucocorticoids to Veterans during 2021-2022, by facility complexity

Specialty	NPI Classification(s)	Provider N	Facility Complexity, N (%)				
			1a	1b	1c	2	3
Primary care	Internal Medicine General Practice	13,278	6,302 (47.5%)	3,452 (25.9%)	2,113 (15.9%)	857 (6.5%)	1,013 (7.6%)
Family Medicine	Family Medicine	5,474	2,129 (38.9%)	1,168 (21.3%)	1,096 (20.0%)	633 (11.6%)	674 (12.3%)

Table V: Providers prescribing glucocorticoids to Veterans during 2021-2022, by facility region

Specialty	NPI Classification(s)	Provider N	Region, N (%)				
			Continental	Midwest	N. Atlantic	Southeast	Pacific
Primary care	Internal Medicine General Practice	13,278	2,319 (17.5%)	3,536 (26.6%)	2,731 (20.1%)	2,620 (19.7%)	2,311 (17.4%)
Family Medicine	Family Medicine	5,474	1,314 (24.0%)	1,336 (24.4%)	915 (16.7%)	1,205 (22.0%)	807 (14.7%)

Table VI: Providers prescribing glucocorticoids to Veterans during 2021-2022, by facility rurality

Specialty	NPI Classification(s)	Provider N	Rurality, N (%)	
			Urban	Rural
Primary care	Internal Medicine General Practice	13,278	12,836 (96.6%)	681 (5.1%)
Family Medicine	Family Medicine	5,474	5072 (92.6%)	505 (9.2%)

c. Also, did the authors consider looking at differences in prescribing patterns among MDs/DOs vs advanced practice providers? This could yield some interesting patterns and could inform dissemination of the findings.

R7c. We agree this would be an interesting and relevant addition to the manuscript. Unfortunately, the provider NPI and NUCC taxonomy system we used to define provider specialty does not reliably allow within-specialty differentiation of MD/DO providers from other provider designations (e.g. NP, PA, etc). We have added this to our limitations section (page 12):

“...inability to differentiate training type within specialty (e.g. physician, nurse practitioner, physician assistant).”

d. In the discussion, the authors report the study supports that steroid use may be stabilizing, but I do not feel the data support this, as the prescribing rate of steroids they report is nearly 50% higher than the rate reported in claims in the prior decade.

R7d. Per the data available, annual rates of glucocorticoid prescribing in the United States increased five-fold between the 1999-2008 estimate of 1.2%, and the 2009-18 estimates of 6.4-7.7%. Rates then increased less than twofold between the 2009-18 estimates and the 2021-22 estimates presented here. While rates of use are indeed rising across this period, the rate of change is falling, suggesting use patterns may be stabilizing. We have reorganized the paragraph containing this observation and clarified its wording as follows (page 10)

“We found a mean glucocorticoid prescribing rate of 12.1% between 2021-2022, compared to annual rates of 6.4% - 7.7% reported in commercial claims between 2009-2018, and an overall rate of 1.2% by NHANES participants between 1999-2008. While prescribing increased steadily over this period, these trends suggest the rate of increase may be stabilizing.”

Q8) In Figures 1 and 2 the numbering of the X axis are too small to read. If the authors are striving to demonstrating variability across VA Medical Centers, I don't think they need to include these specific facility numbers in the X axis – the plot speaks for itself.

R8. We have revised Figures 1 and 2 to remove the facility designations.

Reviewer #3

This study provides a good descriptive accounting on glucocorticoid use using stratified analysis. I would recommend applying a regression model to quantify the contribution of the multiple variables and the relationships between the variables to strengthen the study.

Overall, this study provides evidence to the impact glucocorticoid use has on the population and may help provide support for "glucocorticoid stewardship" as the authors note model similar to antibiotic and opiate prescribing.

Thank you for your comments!

Reviewer #1

All my comments and suggestions were taken into consideration in the authors' response letter and the changes were made on revised manuscript and supplemental materials.

We thank the reviewer for their helpful comments, they were instrumental in improving the manuscript

Reviewer #2

This is a well-written study that effectively leverages the rich data available for the millions of Veterans that VA services. It is motivated by the potential harms associated with glucocorticoid overprescribing and the relatively novel concept of glucocorticoid stewardship. The authors have been thoughtful in their responses to the prior reviewer comments. They have further justified and clarified a number of points and have enhanced the discussion by more thoroughly exploring the implications of their work.

We thank the reviewer for their thoughtful feedback on our manuscript, and appreciate their kind words!

Reviewer #3

I am satisfied with the responses to all reviewers comments.

We thank the reviewer for their input and help with this manuscript